# Hemp (*Cannabis sativa* L.) Seed Protein–EGCG Conjugates: Covalent Bonding and Functional Research

**DOI:** 10.3390/foods10071618

**Published:** 2021-07-13

**Authors:** Xin-Hui Pang, Yang Yang, Xin Bian, Bing Wang, Li-Kun Ren, Lin-Lin Liu, De-Hui Yu, Jing Yang, Jing-Chun Guo, Lei Wang, Xiu-Min Zhang, Han-Song Yu, Na Zhang

**Affiliations:** 1Key Laboratory of Food Science and Engineering of Heilongjiang Province, College of Food Engineering, Harbin University of Commerce, Songbei District, Harbin 150076, China; pxh10010@163.com (X.-H.P.); foodyangyang@163.com (Y.Y.); bianbian1225@163.com (X.B.); iceking85@163.com (B.W.); likun931006@163.com (L.-K.R.); liulinlin721@163.com (L.-L.L.); yudehui1213@163.com (D.-H.Y.); YJYJ6620@163.com (J.Y.); 2Heilongjiang Academy of Sciences, Harbin 150000, China; guojingchun@163.com (J.-C.G.); wleileiyu@163.com (L.W.); 3Beijing Academy of Food Sciences, Beijing 100068, China; zxmchnfood@163.com; 4College of Food Science and Engineering, Jilin Agricultural University, Changchun 130118, China; 5Division of Soybean Processing, Soybean Research & Development Center, Chinese Agricultural Research System, Changchun 130118, China

**Keywords:** hemp protein isolate, (−)-epigallocatechin gallate, covalent binding, structural changes, emulsifying properties

## Abstract

In order to make HPI have a wide application prospect in the food industry, we used EGCG to modify HPI. In this study, we prepared different concentrations (1, 2, 3, 4, and 5 mM) of (−)-epigallocatechin gallate (EGCG) covalently linked to HPI and use methods such as particle size analysis, circular dichroism (CD), and three-dimensional fluorescence spectroscopy to study the changes in the structure and functional properties of HPI after being covalently combined with EGCG. The particle size data indicated that the covalent HPI-EGCG complex was larger than native HPI, and the particle size was mainly distributed at about 200 μm. CD and three-dimensional fluorescence spectroscopy analyses showed that the conformation of the protein was changed by conjugation with EGCG. The β-sheet content decreased from 82.79% to 66.67% after EGCG bound to the protein, and the hydrophobic groups inside the protein were exposed, which increased the hydrophobicity of the protein and changed its conformation. After HPI and 1 mM of EGCG were covalently bonded, the solubility and emulsifying properties of the covalent complex were improved compared with native HPI. These results indicated that HPI-EGCG conjugates can be added in some foods.

## 1. Introduction

Hemp (*Cannabis sativa* L.) is a widely cultivated plant, and has an especially important role in industry. It is divided into two types according to the 9-tetrahydrocannabinol (THC) content: industrial hemp and drug hemp [1]. A 0.3% THC standard has been established by the European Union for this classification. If the THC is less than 0.3%, industrial hemp is allowed to be grown in China and Canada. Hemp is grown for industrial use and harvested for its fiber, seeds, and oil. Hemp seeds are rich in phytosterols, omega-3, and omega-6 essential fatty acids and protein (approximately 25% of dry weight). Moreover, there are many kinds of essential amino acids, and all of the essential amino acids needed by the human body are contained, with relatively high glutamic acid and arginine content [2,3]. Histidine is an essential amino acid for infants less than eight months old, and it plays an important role in the prevention of cardiovascular diseases in middle-aged and elderly people and in the growth of children. Hemp seeds are considered to be a good source of high-quality protein suitable for the elderly and infants. Therefore, they have been used in the production of various foods with high nutritional value. Importantly, the emergence of hemp varieties with low THC content has increased the utilization of hemp in food production [4]. Hemp seeds contain more than 30% oil and 25% protein. Currently, hemp is mainly used for the extraction of hemp oil because of the high oil content in the seeds [5]. However, compared with other plant proteins, hemp protein has low solubility, which is attributed to the aggregation of edestin (11S globulin) at pH values lower than 7.0. Therefore, hemp protein needs to be modified to increase its range of applications.

In recent years, various chemical modification methods, such as phosphorylation, glycosylation, deamidation, and succinylation, have been proven to be effective in improving the functional properties of proteins [6]. Phosphorylation, deamidation, and succinylation methods all result in chemical residues, which lead to a decrease in the nutritional value of the protein. The Maillard reaction is the main method of glycosylation, but it is difficult to control. Overreaction affects the flavor and quality of the products. Therefore, some researchers have proposed the use of polyphenols to modify proteins. In recent years, many researchers have begun to study the interaction between polyphenols and proteins. Proteins and polyphenols can interact through both non-covalent and covalent bonds. Non-covalent interactions are reversible, whereas covalent interactions are irreversible [7]. There are two types of methods used to form protein–polyphenol conjugates: non-enzymatic (alkaline and free radical reactions) and enzymatic (polyphenol oxidase, laccase, and tyrosinase) [8]. The mechanism of the alkaline reaction involves the oxidation of phenolic compounds to quinone compounds in an alkaline solution [9]. Quinone compounds usually react further with nucleophilic amino acid residues in the protein chain. Wei conducted research on the covalent complex of sodium caseinate, β-lactoglobulin, lactoferrin, and α-whey protein with EGCG and found that the covalent complex significantly improved the stability of the β-carotene emulsion [10]. Some researchers have also covalently combined phenolic compounds with flaxseed protein isolate (FPI). They found that the covalent complex of polyphenols and has a higher emulsifying ability than FPI [11]. When He studied the covalent binding of ovalbumin (OVA) and EGCG, he also found that the emulsification of OVA was improved [12]. The addition of phenol allows the protein to acquire phenolic hydroxyl groups, thereby changing the functional properties of the protein.

(−)-Epigallocatechin gallate (EGCG) is the main component of polyphenols and catechins in green tea [13]. EGCG has beneficial physiological activities, such as protecting against free radical DNA damage, antioxidative effects [14], inhibiting tumor growth, reducing serum low-density lipoprotein and cholesterol levels, improving vascular proliferation, and protecting against cardiovascular diseases [15].

Research on polyphenols and proteins has mainly focused on non-covalent interactions, but these interactions are reversible and unstable. Therefore, we chose to study covalent interactions between polyphenols and proteins. The covalent complexes formed by hemp protein isolate (HPI) and EGCG were used to investigate these interactions. Zeta potential values, CD spectra, and three-dimensional fluorescence spectra were used to explore the structure of the HPI-EGCG complex. The functional characteristics of the covalent complex were determined by analyzing the reactive groups, polyphenol content, the size of the particles, and other relevant indicators. The results of this research provide some theories and provide a research basis for future researchers for the application of the HPI-EGCG complex in the food industry as an emulsifier.

## 2. Materials and Methods

### 2.1. Experimental Materials

Hemp was purchased from the Heilongjiang Academy of Sciences (Harbin, China). Hemp protein samples were prepared by alkali-soluble acid precipitation. EGCG, *o*-phthaldialdehyde (OPA), and 5,5′-dithio-2-nitrobenzoic acid (DTNB) were purchased from Solarbio Life Science Co., Ltd. (Beijing, China). Other reagents were of analytical grade and were purchased from Tianjin Kermel Chemical Reagent Co., Ltd. (Tianjin, China).

### 2.2. Extraction of HPI

HPI was extracted according to Tang’s method, with some changes [16]. Degreased powder was dissolved in 20 volumes of distilled water, and the extraction solution was adjusted to a pH of 9.0 generally with 2 mol/L of NaOH, stirred at 50 °C for 1 h, and the centrifugal force was set to 10,000× *g* at 4 °C. The centrifugal time was 20 min to yield a clear liquid. The pH of the clear liquid was adjusted to 4.8 with 2 mol/L of HCl, the centrifugal force was set to 8000× *g* at 4 °C, the centrifugal time was 20 min, and the liquid was cooled overnight. The supernatant was discarded and the remaining precipitate was washed with distilled water three times to remove the salt. The resulting precipitate was then dissolved in distilled water, adjusted to a pH of 7.0, placed in a low-temperature refrigerator for pre-freezing for 24 h, and freeze-dried. The protein content of the freeze-dried powder, as determined by the Kjeldahl method, was 93.7% (N × 6.25).

### 2.3. Preparation of Covalent HPI-EGCG Complexes

Covalent HPI-EGCG complexes were prepared using a previously described method [17], with slight modifications. HPI (0.5 g) was dissolved in 50 mL of distilled water. Different concentrations of the EGCG mother liquor were prepared, and the HPI and EGCG solution was mixed in a volume ratio of 1:1 to ensure that the concentration of EGCG in the mixed solution was 0, 1, 2, 3, 4, or 5 mM. The solution was adjusted to a pH of 9.0 and magnetically stirred for 24 h under an aerobic condition at 25 °C and then dialyzed. The HPI-EGCG conjugate was obtained by freeze-drying.

### 2.4. Determination of Sulfhydryl Content

The determination of sulfhydryl content was based on the method of previous researchers [18], with slight modifications. Ellman reagent was prepared with 4 mg of DTNB and 1 mL of Tris-glycine. Tris-glycine (contained 8 M of urea) buffer was used to prepare an HPI solution at a concentration of 3 mg/mL. Fifty microliters of Ellman reagent was then added to 5 mL of the HPI solution and reacted for 1 h in a dark environment. The absorbance value under ultraviolet light was 412 nm, and the sulfhydryl content was calculated using the following formula:sulfhydryl content (μmol/g protein) = (73.53 × A_412_ × D)/C(1)
where A_412_ is the absorbance at 412 nm, D is the dilution coefficient, and C is the protein concentration of the HPI sample (mg/mL).

### 2.5. Determination of Free Amino Group Content

The free amino group content of HPI and the HPI-EGCG complexes was determined using the *o*-phthaldialdehyde (OPA) method [19]; 80 mg of OPA powder was accurately weighted, added to 4 mL of methanol solution, and mixed to prepare OPA reagent. Then, 50 mL was added to 0.1 M of borax solution, 200 μL of β-mercaptoethanol, and 5 mL of sodium lauryl sulfate solution (20%, *w*/*w*). The resulting solution was transferred to a volumetric flask and made up to 100 mL with distilled water. Two hundred microliters of the sample solution, at a concentration of 1 mg/mL, was mixed with 4 mL of the OPA reagent and vortexed to obtain a homogeneous solution. After protecting from light for 2 h, the spectrophotometer was set to 340 nm and the absorbance of the solution was measured. l-leucine solutions of different concentrations were used to construct a standard curve of the absorbance value vs. the l-leucine concentration. The content of free amino groups was calculated with the obtained absorbance value through the standard curve.

### 2.6. Determination of EGCG Content

Total phenol equivalents were determined in the HPI and HPI-EGCG complex samples [20]. Folin–Ciocalteu reagent (2.5 mL, 0.2 N) was added to 0.5 mL of HPI-EGCG complex solution and mixed thoroughly using a wandering oscillator. After reacting for 5 min, 2 mL of an Na_2_CO_3_ solution (7.5%, *w*/*v*) was added, and the mixture was shaken for 30 s and reacted for 2 h in the dark. Absorbance was then measured at 760 nm, usually with distilled water as a blank. EGCG standard solutions of different concentrations were used to construct a standard curve of absorbance vs. EGCG concentration. The EGCG content of the sample was then calculated according to the standard curve, and the final result was expressed as μmol EGCG/g protein.

### 2.7. Zeta Potential and Particle Size Determination

According to the method used by previous researchers [21], the zeta potential and particle size of the covalent complex were measured, with some modifications.

The average particle size distribution of each sample was determined, and a sample solution was prepared with a concentration of 0.2 mg/mL using a Malvern particle size analyzer (Malvern Panalytical, Malvern, UK). The samples were centrifuged at 4000× *g* for 15 min and passed through a 0.45 μm cellulose acetate filter membrane, and the particle size was then measured at room temperature.

A Nano ZS90 zetasizer was used to determine the zeta potential, according to the method used by previous researchers [22], with slight modifications. Each sample was measured six times, and the average value was calculated as the final result.

### 2.8. Circular Dichroism Analysis

Far-ultraviolet circular dichroism (CD) can analyze the changes in the secondary structure of the HPI-EGCG complex [23]. A sample solution with a protein concentration of 0.5 mg/mL was prepared in distilled water. Far-ultraviolet CD measurements were performed in a 1 mm diameter quartz cuvette using an MOS-450 spectrophotometer (French Bio-Logic SAS, Seyssinet-Pariset, France). The following parameters were used: spectral scan range, 190–250 nm; scan resolution, 1 nm; scan rate, 100 nm/min; and general band width, 2.0 nm. Samples were scanned nine times and the average value was calculated.

### 2.9. Three-Dimensional Fluorescence Spectroscopy

The change in fluorescence intensity of the samples was measured using an F-6000 fluorometer spectrophotometer (Hitachi Ltd., Tokyo, Japan) [24]. A 1 mg/mL sample was prepared, and the samples were placed in a quartz cuvette for measurement. The excitation wavelength (λ_Ex_) was set at 200–350 nm. Continuous scanning of the three-dimensional fluorescence spectrum was performed at an emission wavelength (λ_Em_) ranging from 200 to 500 nm, with an initial λ_Ex_ of 200 nm, a scanning speed of 3000 nm/min, and a slit width of 5 nm.

### 2.10. HPI-EGCG Conjugate Properties

#### 2.10.1. Solubility

The total protein content change of the sample solutions was determined using the Garcia method [25], with slight modifications; 20 mL of the 5.0 mg/mL protein sample was stirred for 1 h at a constant temperature of 25 °C. The samples were then centrifuged at 3000× *g* for 15 min, and the amount of soluble protein was determined using the following formula:SI (%) = W_2_/W_1_ × 100(2)
where SI is the protein solubility, W_1_ is the original protein quantity (g), and W_2_ is the soluble protein quantity (g).

#### 2.10.2. Emulsifying Properties

Emulsification and emulsification stability were determined as previously described [26]. Briefly, 5 mL of soybean salad oil were added to 15 mL of a concentration of 3.0 mg/mL HPI-EGCG conjugate solution at a pH of 7.0 and centrifuged at 9000× *g* for 1 min. A 50 μL sample of the emulsion was removed and mixed with 0.1% (*w*/*v*) sodium dodecyl sulfate. The absorbance of the sample at 500 nm at 0 and 30 min was recorded. The following formulas were used to calculate the Emulsification Activity Index (EAI) and Emulsification Stability Index (ESI):EAI (m^2^/g) = 2 × 2.303 × A_0_ × dilution factor/100,000 × λ × C (3)
ESI (%) = A_30_/A_0_(4)
where the dilution factor is 100, C is the protein concentration (g/mL), λ is the volumetric oil fraction, A_0_ is the absorbance measured at 0 min, and A_30_ is the absorbance measured at 30 min.

### 2.11. Cryo-Scanning Electron Microscopy

A cryogenic scanning electron microscope (JSM-7100; Jeol Europe, Zaventem, Belgium) was used to observe the microstructure of the emulsion. A field emission scanning electron microscope was used to capture still images of the microstructure of the emulsions. The emulsion was added dropwise into the short tube of a low-temperature scanning electron microscope located on the holder and frozen in nitrogen. Subsequently, the short tube was transferred to an ultra-vacuum low-temperature chamber, and the emulsion was broken and etched at −95 °C for 60 s. The broken surface was coated in platinum and then transferred to the freezing stage (−190 °C) of the scanning electron microscope. Images were captured using the microscope control software [27].

### 2.12. Statistical Analysis

Each group of experiments was repeated in triplicate, and the data were analyzed using SPSS 23.0 software (IBM, Armonk, NY, USA). The variance of each treatment group and an independent-sample Student’s *t*-test were used to determine whether there were significant differences between the samples (*p* < 0.05). The results are expressed as mean ± standard deviation (SD). 

## 3. Results and Discussion

### 3.1. Determination of HPI and EGCG Covalent Reactive Groups and EGCG Content

In an alkaline environment, the phenolic hydroxyl group of EGCG is oxidized to produce quinones, which changes the sulfhydryl or amino groups of the protein. C-S or C-N bonds are formed during the covalent bonding of polyphenols and proteins. In this study, an increase or decrease in the content of sulfhydryl and free amino groups concentrations of HPI during the reaction with EGCG and the amount of polyphenol binding in the reaction was measured (Table 1). In the process of covalent bonding of HPI and EGCG, the content of side-chain amino groups in the protein also changed with changes in the amount of EGCG added.

The sulfhydryl contents of HPI and the HPI-EGCG covalent complex were measured within the allowable error range of the method. EGCG was reacted with HPI to form covalent bonds. When the concentration of EGCG continued to rise, the sulfhydryl content of HPI decreased from 28.41 μmol/g HPI to 9.14 μmol/g HPI (Table 1). The process of converting sulfhydryl groups into disulfide bonds can be prevented by urea, and therefore, EGCG can be derived from the decrease in free sulfhydryl content. Covalent bonding occurred with the free sulfhydryl group of HPI; that is, the phenol rings of polyphenols, such as EGCG, can covalently react with the nucleophilic groups of HPI to form C-N and C-S bonds, which illustrates the occurrence of covalent reactions. This is the main reason for the reduction in the number of sulfhydryl groups [28].

The OPA method can sensitively reflect relevant information about the side chain of lysine, the N-terminal residues, and α- or ε-amino groups of proteins, which are the main reactive groups measured by the OPA method. As shown in Table 1, compared with control HPI, there was a downward trend in the content of free amino groups in the covalent complex of HPI and EGCG. When the concentration of EGCG continued to increase, the number of free amino groups in the conjugate greatly decreased. Because free amino groups were quantified in the presence of a 1% concentration sodium dodecyl sulfate, a denaturant that destroys non-covalent protein interactions, we deduced that the decrease in the number of free amino groups may have indicated that the free amino groups of HPI were covalently linked with EGCG. Free amino content groups in HPI were expected to decrease after covalent bonding with EGCG, because the amino acid residues bind to quinone after EGCG oxidation. The ε-amino group contained in lysine was oxidized to form a carbonyl group, and a condensation reaction occurred with the free amino groups, thus reducing the number of amino groups [29].

To further confirm the covalent linkage between HPI and EGCG, the Folin–Ciocalteu method was used to determine the content of EGCG in the solution of the HPI-EGCG conjugate. When determining polyphenol content, HPI samples were pre-measured to have the same protein concentration as the HPI-EGCG samples under the same experimental conditions. Additionally, HPI was used as a blank control to reduce the interference caused by polyphenols. It can be seen from Table 1 that the concentration of EGCG increased and the number of polyphenols increased. However, when the EGCG concentration was 4 mM and 5 mM, the polyphenol content did not significantly change. This implies that, at the concentration of 4 mM, EGCG had reacted with all of the active groups exposed by specific amino acids. It can also be seen from Table 1 that, as more polyphenols were bound, the content of sulfhydryl groups was reduced, as well as the content of free amino groups.

### 3.2. Particle Size Distribution and Zeta Potential Change Analysis

It can be seen from the particle size distribution and zeta potential changes of the HPI-EGCG complexes and HPI samples that, compared with HPI, the covalent HPI-EGCG complex had larger particles (Figure 1). The particle size of the covalently bound protein was mainly about 200 μm. In addition, with an increase in the amount of EGCG added, the height of the peak representing covalent complexes with a particle size of less than 100 μm decreased. The reason for this phenomenon is related to the covalent bonding of EGCG and the protein, as described in the following.

The zeta potential represents the amount of charge on or near the particle surface. The amount of charge is affected by the liquid environment (pH, ionic strength, or presence of surfactants) [30]. The electric potential is related to the size of the interaction between the charged groups, and plays an important role. It can be seen that, as the EGCG content continued to increase, the absolute zeta potential values of the samples increased (Figure 1). This increase in absolute zeta potential is due to the fact that the EGCG molecule is a negatively charged molecule, and, after binding, the protein is also negatively charged. The dispersion and aggregation of particles are related to their surface charge [31]. The absolute value of the zeta potential becomes larger indicates that the system is relatively stable. In the covalently bound samples, the oxidation and cross-linking of EGCG formed a stable network structure. Therefore, as the content of EGCG continued to increase, so too did both the zeta value and the size of the protein particles.

### 3.3. CD Spectroscopy

Far ultraviolet CD spectroscopy can analyze the secondary structure changes of HPI and its conjugates (Figure 2). HPI is a globulin. Its far-UV CD spectrum has a specific negative peak in the range of 210–220 nm. In this study, the maximum value of the negative peak shifted from 213 to 208 nm as the EGCG content continued to rise. Table 2 illustrates the changes in the secondary structure of the HPI-EGCG complex. After the protein was covalently bound to EGCG, the maximum negative value showed an increasing trend. That is, the binding of EGCG and HPI reduced the ellipticity of the protein as measured by CD spectroscopy, which indicated that the secondary structure of HPI had changed after EGCG was added. The secondary structural of natural HPI was composed of 1.79% α-helix, 82.79% β-sheet, and 15.50% unordered. After covalent bonding, the β-sheet content decreased, while the α-helix content and random coil content both increased. The decrease in β-sheet content is the structure that determines the flexibility of protein molecules. The increase in their content indicates that the protein chain becomes loose after covalent interaction and the protein is stretched [32]. Liu also observed this change in a study of the covalent interaction between porcine bone protein hydrolysate and rutin, in which the β-sheet and β-turn content decreased and the α-helix and random helix content increased [33]. In addition, the β-sheet content affects the internal hydrophobic groups of the protein. After covalent bonding, the β-sheet content of the complex was reduced, allowing the internal hydrophobic groups to be exposed, which increased the hydrophobicity of the protein. This change is a sign of a change in protein conformation [34]. Furthermore, Damodaran’s research also found that α-helix and random coil structures are more flexible than β-sheet structures, which indicates that covalent bonding increases the flexibility of the secondary conformation of HPI [35]. These changes are brought about by the binding of hydroxyl groups at the protein hydroxyl, amino, or sulfhydryl sites in the reaction system, which significantly changes the secondary structure of the protein.

### 3.4. Three-Dimensional Fluorescence Spectroscopy

Three-dimensional fluorescence spectroscopy was used to determine the effect of EGCG on the structure of HPI. Peak (a) in the three-dimensional contour map represents Raman scattering. The formation of the complex between EGCG and HPI increased the particle size and enhanced the light scattering effect, thereby increasing the fluorescence intensity of peak (a). Peak 1 shows that the characteristic peaks produced by tryptophan and tyrosine turned blue after the addition of EGCG, which indicated that the fluorescence intensity of the protein decreased after EGCG was combined with HPI [36]. It can be seen from that, as the amount of EGCG increased, the color of the characteristic peak became lighter as the contour line became thinner, and the fluorescence intensity of the protein decreased (Figure 3). The reason for this phenomenon is the covalent interaction between the Trp or Tyr residues in HPI and the quinone formed by the oxidation of polyphenols [37]. Therefore, the fluorescence intensity of HPI decreased after combining with EGCG, which indicated a change in the conformation of HPI. These results confirmed the covalent bonding between EGCG and HPI. Under aerobic conditions, phenol is easily oxidized to quinone, and further oxidation causes a dimerization reaction. Therefore, it is oxidized under aerobic conditions to form *o*-quinone, which can be reduced by the amino groups of the protein side chain, thereby forming a covalent C-N or C-S bond between the HPI and EGCG. These C-N and C-S bonds can form dimers through cross-linking reactions to provide new polymers.

### 3.5. Functional Properties of SPI-EGCG Conjugates

#### 3.5.1. Solubility

The solubility of HPI refers to the percentage of soluble protein in HPI under a specific environment. The solubility of a protein affects its ability to emulsify. HPI had good solubility at 1 mM in (Figure 4). Compared with native HPI, the solubility of HPI-EGCG increased from 39.4% to 50.6%. Thus, the conjugation of HPI with EGCG markedly altered its solubility. The solubility of the conjugate increased at the optimal polyphenol loading (1 mM). The conjugation of EGCG enhanced the solubility of HPI, probably due to EGCG and HPI molecules combining to react, which changed the net charge of the protein molecules. Negatively charged EGCG bound to the hydrophobic area on the HPI surface. On the one hand, it increased the surface electronegativity of the HPI particles due to electrostatic repulsion between the particles, which promoted the dispersion of the protein particles in the aqueous solution to avoid coagulation. On the other hand, the hydrophilic tail of the phenolic hydroxyl group enhanced the interaction between HPI and water molecules [38]. Solubility showed a downward trend at other concentrations (2 mM, 3 mM, 4 mM, and 5 mM) due to excessive polyphenols, which caused protein molecules and polyphenols to form aggregates. The low solubility of such aggregates increased the turbidity of a solution [39]. The result we got is consistent with the result of Quan’s study [40], who found that when the concentration of polyphenols exceeds the concentration of protein binding sites, proteins and polyphenols form aggregates, thereby greatly reducing their solubility. Therefore, the covalent bonding of EGCG and HPI resulted in a substantial change in the solubility of HPI.

#### 3.5.2. Emulsifying Properties of HPI-EGCG Conjugates

The EAI reflects the interfacial tension of protein droplets at the oil–water interface and the ability to stabilize the emulsion. It is determined by protein–protein and protein–lipid interactions. The ESI refers to the stable strength of the emulsion in the dispersion [41]. It can be seen that, except for the 1 mM EGCG complex, the EAI of the HPI-EGCG complexes was lower than that of the control in (Figure 5). Specifically, the ESI value first increased and then decreased with increasing EGCG concentration, reaching a maximum when EGCG was added at 1 mM. Protein–protein and protein-lipid interaction affects the emulsifying properties of proteins and plays an important role in the middle [42]. The combination of protein and the appropriate concentration of EGCG may change the protein–protein interactions between molecules, thereby reducing the interfacial tension of the oil–water interface [43]. The improvement in emulsification performance may be due to the change in the flexibility of HPI after combining with EGCG. The solubility of the protein is improved and the surface hydrophobicity is increased, making protein particles more stable at the oil–water interface [44]. The emulsification of lactoferrin has also been shown to improve when covalently linked with EGCG.

### 3.6. Cryo-Scanning Electron Microscopy Microstructure

Cryo-scanning electron microscopy images can be seen for HPI and covalent HPI-EGCG complex emulsions in Figure 6. The oil droplets were uniformly attached to the surface of the HPI-EGCG molecules, which had a micro-spherical structure [45]. This result is consistent with the results of a previous study showing that when anthocyanins are covalently bound to HPI, they can disrupt the protein peptide chain and enhance the interactions between droplets to form an emulsion [46]. It can be seen that the native state of the HPI emulsion is flocculated, in which fat globules flocculate without breaking the membrane. The droplets of the emulsion continued to aggregate, and as aggregation increased, flocculation also increased, and emulsification occurred more rapidly [47]. After HPI and EGCG were covalently bound, the flocculation phenomenon in the emulsion was suppressed, with the greatest uniformity seen when EGCG was at 1 mM. This shows that the negative charge of EGCG helped stabilize the emulsion and make it more uniform [48].

## 4. Conclusions

In this study, a protein–polyphenol covalent complex was formed by combining HPI with different concentrations of a polyphenol. The addition of EGCG caused changes in the structure of HPI and improved functional properties. After HPI is combined with 1 mM of EGCG, the emulsification and solubility of the covalent complex are improved. These data provide the theoretical basis for the application of polyphenols and protein covalent complex food processing emulsifier.

## Figures and Tables

**Figure 1 foods-10-01618-f001:**
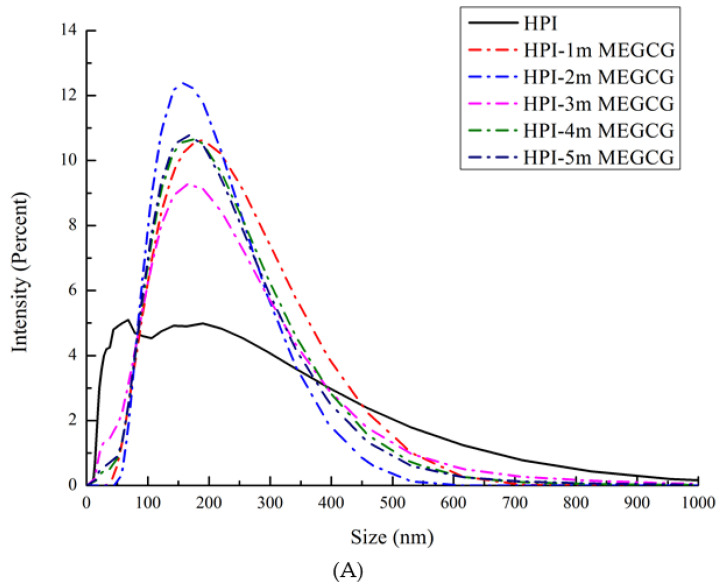
Size distribution (**A**) of HPI and covalent HPI-EGCG complexes at EGCG concentrations of 1, 2, 3, 4, and 5 mM. Zeta potential (**B**) of HPI and covalent HPI-EGCG complexes at EGCG concentrations of 1, 2, 3, 4, and 5 mM.

**Figure 2 foods-10-01618-f002:**
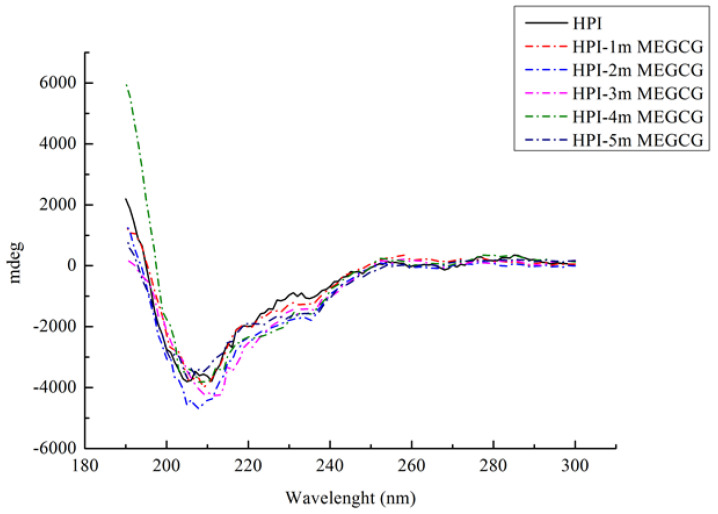
Far-UV CD spectroscopy of HPI and HPI-EGCG conjugates.

**Figure 3 foods-10-01618-f003:**
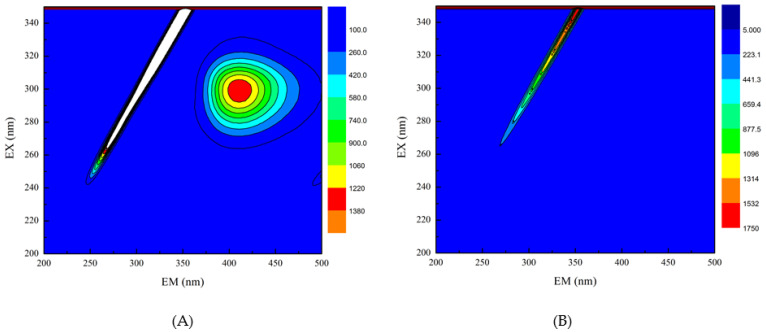
Three-dimensional fluorescence spectra of HPI and covalent HPI-EGCG complexes. (**A**) HPI, (**B**) HPI-1 Mm EGCG, (**C**) HPI-2 mM EGCG, (**D**) HPI-3 mM EGCG, (**E**) HPI-4 mM EGCG, and (**F**) HPI-5 mM EGCG, all suspended in H_2_O (Ex is the excitation wavelength and Em is the emission wavelength).

**Figure 4 foods-10-01618-f004:**
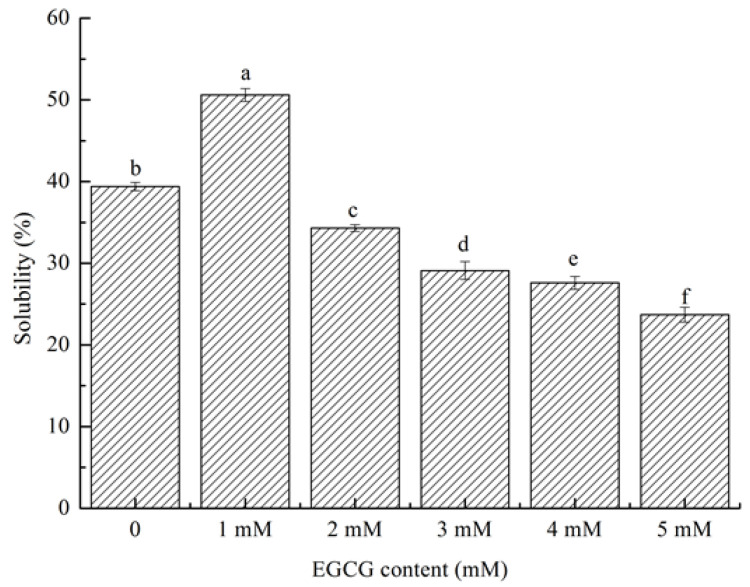
(a-f) Solubility of HPI and covalent HPI-EGCG complexes at EGCG concentrations of 1, 2, 3, 4, and 5 mM.

**Figure 5 foods-10-01618-f005:**
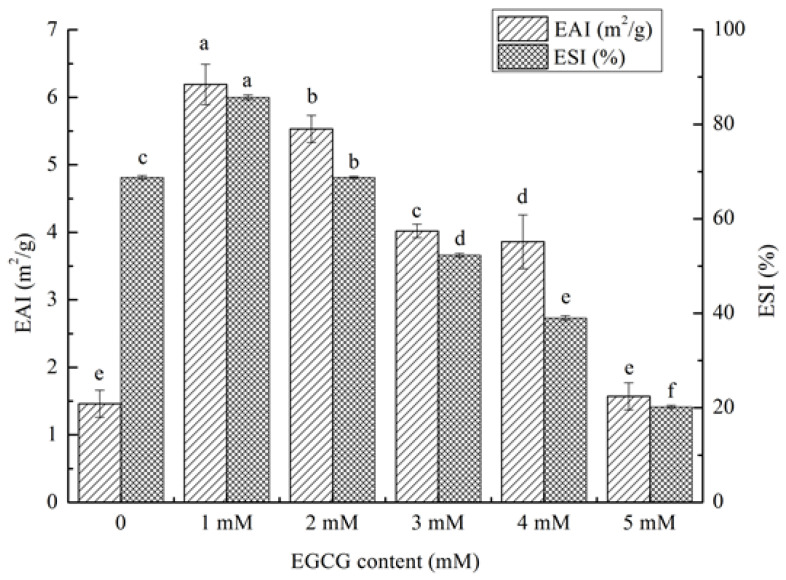
(a–f) Emulsifying properties of HPI and covalent HPI-EGCG complexes at EGCG concentrations of 1, 2, 3, 4, and 5 mM (EAI is the Emulsifying Activity Index and ESI is the Emulsifying Stability Index).

**Figure 6 foods-10-01618-f006:**
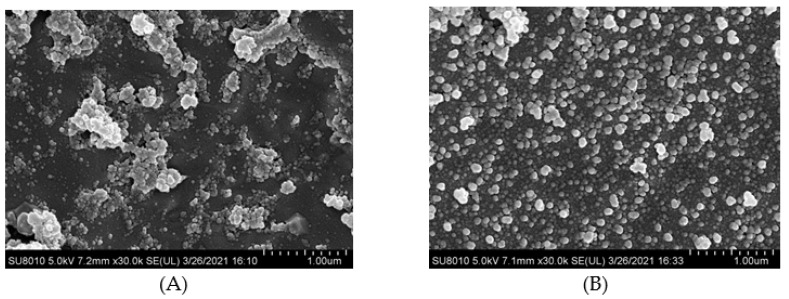
Cryo-scanning electron microscope images of HPI and covalent HPI-EGCG complexes at EGCG concentrations of 1, 2, 3, 4, and 5 mM. (**A**) HPI, (**B**) HPI-1 Mm EGCG, (**C**) HPI-2 mM EGCG, (**D**) HPI-3 mM EGCG, (**E**) HPI-4 mM EGCG, and (**F**) HPI-5 mM EGCG.

**Table 1 foods-10-01618-t001:** The sulfhydryl group, free amino group, and EGCG concentrations in HPI-EGCG conjugates.

Sample	Sulfhydryl Groups (μmol/g Protein)	Free Amino Groups (μmol/g Protein)	EGCG Content (μmol/g Protein)
HPI	28.41 ± 0.06 ^a^	225.33 ± 0.7 ^a^	—
HPI-1 mM EGCG	17.48 ± 0.13 ^b^	202.02 ± 5.1 ^b^	23.08 ± 1.1 ^c^
HPI-2 mM EGCG	14.58 ± 0.14 ^c^	196.02 ± 1.9 ^c^	55.79 ± 1.8 ^c^
HPI-3 mM EGCG	12.53 ± 0.08 ^d^	173.39 ± 1.2 ^d^	87.38 ± 3.0 ^bc^
HPI-4 mM EGCG	9.14 ± 0.01 ^e^	84.39 ± 2.3 ^e^	100.15 ± 0.9 ^b^
HPI-5 mM EGCG	9.96 ± 0.03 ^e^	65.29 ± 2.8 ^f^	109.80 ± 3.3 ^a^

Values are expressed as the mean ± SD. Different letters in the same column indicate significant differences (*p* < 0.05).

**Table 2 foods-10-01618-t002:** Secondary structure contents of HPI and HPI–EGCG conjugates.

Sample	α-Helix (%)	β-Sheet (%)	β-Turn (%)	Random Coil (%)
HPI	1.79	82.79	0	15.50
HPI-1mMEGCG	1.87	71.16	0	27.30
HPI-2mMEGCG	3.33	66.86	2.33	27.76
HPI-3mMEGCG	3.90	66.67	3.40	25.53
HPI-4mMEGCG	5.41	69.60	0	24.43
HPI-5mMEGCG	1.94	70.67	1.43	25.67

## Data Availability

Research data are not shared.

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
