# Peer review of "Hemp (Cannabis sativa L.) Seed Protein–EGCG Conjugates: Covalent Bonding and Functional Research"

_foods, 2021, doi:10.3390/foods10071618_

Round 1
Reviewer 1 Report
The paper entitled „Hemp (Cannabis sativa L.) Seed Protein–Epigallocatechin Gallate Conjugates: Covalent Bonding and Functional Characteristics” regards a study on covalent complex between EGCG and hemp protein isolate, its structure and functional properties. The topic is interesting and worth researching. Generally, the paper is written in clear language and is organized properly. The methods are correctly selected for the purpose of the research. I recommend a minor revision. I have anyway some comments and question to the authors:
- Introduction is quite general. It would be more interesting if you discuss some previous works on covalent complex formation between specific proteins and polyphenols; what could be the potential applications of EGCG-HPI complex in food industry?
- The purpose of the research should be concise and clearly written. In this case, it is too long and convoluted.
- In lines 231-233 you write: “With increasing EGCG concentrations, the sulfhydryl content of HPI decreased from 8.27 μmol/g HPI to 0.17 μmol/g HPI (Table 1).” I can’t see these data in table 1.
- Figure 3 – I am not a big fan of such presentation of results, because they are illegible to me. The caption under the figure is not detailed enough. Please write down what is in picture A, B etc. I understand that the colored bars indicate the intensity of the fluorescence. This information should be included in the caption.
- Figure 5 - explain the abbreviations EAI and ESI in the caption under the figure.
Some minor points:
- According to instruction for authors you should place reference numbers throughout the manuscript in square brackets [ ]. References are written inconsistently with the instructions.
- Line 348 and 375, 393 – remove parentheses (as well as in other places where you refer directly to the graph or table).
- Line 63 – you can remove “this reaction”
- Line 103 – “Degreasing” or Degreased?
Author Response
30-Jun-2021
Dear Editors and Reviewers:
Thank you for your letter and the comments on reviewers concerning our manuscript entitled “Hemp (Cannabis sativa L.) seed protein–EGCG conjugates: Covalent bonding and functional research” (ID: foods-1268419). Those comments are all valuable and extremely helpful for revising and improving our paper, and they also have an important guiding significance on our researches. We have studied your comments carefully and made corrections and revised portions are marked by red . We sincerely hope this manuscript will be finally acceptable to be published on the Foods.
Main corrections in the paper and the responses to the reviewer’s comments are as follows:
Responses to the reviewer’s comments:
Reviewer #1:
1 Introduction is quite general. It would be more interesting if you discuss some previous works on covalent complex formation between specific proteins and polyphenols; what could be the potential applications of EGCG-HPI complex in food industry?
Response: Thank you for your reminder. The Introduction of the manuscript was corrected, and kept the mainly propose of this manuscript. We supplemented some previous works on covalent complex formation between specific proteins and polyphenols. The supplemented content was:“Wei conducted research on the covalent complex of sodium caseinate, β-lactoglobulin, lactoferrin and α-whey protein with EGCG and found that the covalent complex significantly improved the stability of the β-carotene emulsion[1]. Some researchers have also covalently combined phenolic compounds with flaxseed protein, they found that the covalent complex of polyphenols and flaxseed protein has a higher emulsifying ability than flaxseed protein[2]. When He studied the covalent binding of ovalbumin and EGCG, he also found that the emulsification of ovalbumin was improved[3].” The potential applications was that EGCG improves the emulsification of HPI, this covalent complex can be used in food formulations as emulsifiers .The revised Introduction was shown on Line 80-87 and Line 105 of the revised manuscript.
References
[1] Wei, Z., Yang, W., Fan, R., Yuan, F., & Gao, Y. Evaluation of structural and functional properties of protein–EGCG complexes and their ability of stabilizing a model β-carotene emulsion. Food Hydrocolloids 2015, 45, 337–350.
[2] Phama L. B., Wang B., Zisuc B., Adhikaria B. Covalent modification of flaxseed protein isolate by phenolic compounds and the structure and functional properties of the adducts. Food Chemistry 2019, 293, 463-471.
[3] He W. Y., Xu H. X., Lu Y. Q., Zhang T. T., Li S. M., Lin X., Xu B. Q., Wu X. L. Function, digestibility and allergenicity assessment of ovalbumin–EGCG conjugates. Journal of Functional Foods 2019, 61, 103490.
2 The purpose of the research should be concise and clearly written. In this case, it is too long and convoluted.
Response: Thank you for your suggestions, all of which are very important. We re-edited the purpose, and kept the mainly propose of this manuscript. The modified content was: “In order to make HPI have a wide application prospect in the food industry, we used EGCG to modify HPI.” The revised purpose of the research was shown on Line 24-25 of the revised manuscript.
3 In lines 231-233 you write: “With increasing EGCG concentrations, the sulfhydryl content of HPI decreased from 8.27 μmol/g HPI to 0.17 μmol/g HPI (Table 1).” I can’t see these data in table 1.
Response: Thank you for pointing out this problem. We have modified the sentence, and the revised content was “When the concentration of EGCG continues to rise, the sulfhydryl content of HPI decreased from 28.41 μmol/g HPI to 9.14 μmol/g HPI”. The Lines 251-252 of the manuscript was corrected, please refer to the red of Lines 251-252 in the revised manuscript.
4 Figure 3 – I am not a big fan of such presentation of results, because they are illegible to me. The caption under the figure is not detailed enough. Please write down what is in picture A, B etc. I understand that the colored bars indicate the intensity of the fluorescence. This information should be included in the caption.
Response: Thank you for pointing out this problem. According to the comments, we have modified Figure 3 to make the data more intuitive. The revised figure was shown below. The caption under the figure was corrected, please refer to the red of Lines 403-405 in the revised manuscript.
(A) (B)
(C) (D)
(E) (F)
Fig. 3. Three-dimensional fluorescence spectra of HPI and covalent HPI-EGCG complexes (A)HPI, (B)HPI-1MmEGCG, (C)HPI-2mMEGCG, (D) HPI-3mMEGCG, (E) HPI-4mMEGCG, (F) HPI-5mMEGCG. (Ex is the excitation wavelength, Em is the Emission Wavelength).
5 Figure 5 - explain the abbreviations EAI and ESI in the caption under the figure.
Response: The abbreviations EAI and ESI in the caption under the figure was explained, EAI is the Emulsifying Activity Index, ESI is the Emulsifying Stability Index, please refer to the red of Lines 472 in the revised manuscript.
6 According to instruction for authors you should place reference numbers throughout the manuscript in square brackets [ ]. References are written inconsistently with the instructions.
Response: Thank you for your suggestion. We have placed reference numbers throughout the manuscript in square brackets [ ], and the full text was checked.
7 Line 348 and 375, 393 – remove parentheses (as well as in other places where you refer directly to the graph or table).
Response: Thank you very much, we have removed the parentheses.
8 Line 63 – you can remove “this reaction”
Response: Thank you for pointing out this problem, Line 63 “this reaction” have been removed.
9 Line 103 – “Degreasing” or Degreased?
Response: Thank you for pointing out this problem, “Degreasing” was modified to “Degreased”, please refer to the red of Line 103 in the revised manuscript.
The above is our revised responses based on suggestions of the editor and reviewers.
Thank you again for your suggestions. We hope we can learn more from you.
We look forward to your reply!
Sincerely,
Prof. Na
Harbin University of Commerce

Reviewer 2 Report
This work approaches the formation of hemp seed protein-EGCG conjugates to improve the functional properties of hemp protein isolates, and some interesting results are provided. The authors performed a wide range of experiments to reach to the conclusions.
There is a major comment that I would like the authors to clarify. The polyphenol content in native sample should have been given. Could it have played any role in association with EGCG?
The solubility experiment looks a little odd; dissolving 1g sample in 100 ml H2O. Could the authors double-check?
Some minor comments:
Line 1 “Hemp is a native crop of China”. Reference?
3D FL-Spectra on line 340; Fig. 3C has a different formatting (scale tics) compared to other figures in the groups.
Author Response
30-Jun-2021
Dear Editors and Reviewers:
Thank you for your letter and the comments on reviewers concerning our manuscript entitled “Hemp (Cannabis sativa L.) seed protein–EGCG conjugates: Covalent bonding and functional research” (ID: foods-1268419). Those comments are all valuable and extremely helpful for revising and improving our paper, and they also have an important guiding significance on our researches. We have studied your comments carefully and made corrections and revised portions are marked by red . We sincerely hope this manuscript will be finally acceptable to be published on the Foods.
Main corrections in the paper and the responses to the reviewer’s comments are as follows:
Responses to the reviewer’s comments:
Reviewer #2:
1 There is a major comment that I would like the authors to clarify. The polyphenol content in native sample should have been given. Could it have played any role in association with EGCG?
Response: Thank you very much for your recognition and suggestion of my article, all of which are very important. Most of the phenolic substances in hemp are present in hemp plants. The phenolic substances in hemp seeds is very small and they are almost insoluble in water, soluble in organic solvents such as ethanol, methanol, ether, benzene, and chloroform. Our HPI is extracted in hemp seeds by aqueous solution, so HPI hardly contains polyphenols.
References:
[1] Wei, Z., Yang, W., Fan, R., Yuan, F., & Gao, Y. Evaluation of structural and functional properties of protein–EGCG complexes and their ability of stabilizing a model β-carotene emulsion. Food Hydrocolloids 2015, 45, 337–350.
2 The solubility experiment looks a little odd; dissolving 1g sample in 100 ml H2O. Could the authors double-check?
Response: Thank you for pointing out our negligence. Corresponding revisions have been made in the revised manuscript, please refer to the red of Line184-185 in the revised manuscript. The solubility experiment was to prove that the combination of HPI and EGCG can improve the functional characteristics of HPI and help the application of HPI in actual production. Accurately weigh 100 mg of the sample and dissolve it in 20 mL of distilled water to prepare a sample solution with a concentration of 5.0 mg/mL. Dissolve it at room temperature for 1 h and centrifuge to determine the protein content in the supernatant.
3 Line 1 “Hemp is a native crop of China”. Reference?
Response: Appreciated your advice very much. What we wrote before was not very rigorous. We have checked the literature and revised this sentence. The revised content is: Hemp(Cannabis sativa L.) is a widely cultivated plant, especially has an important role in industry. Please refer to the red of Line 43-44 in the revised manuscript.
4 3D FL-Spectra on line 340; Fig. 3C has a different formatting (scale tics) compared to other figures in the groups.
Response: Thank you for pointing out the problem. We have modified the format of Figure 3C to make it consistent with other image formats in this group. The revised Figure 3 is as follows.
(A) (B)
(C) (D)
(E) (F)
Fig. 3. Three-dimensional fluorescence spectra of HPI and covalent HPI-EGCG complexes. (A)HPI, (B)HPI-1MmEGCG, (C)HPI-2mMEGCG, (D) HPI-3mMEGCG, (E) HPI-4mMEGCG, (F) HPI-5mMEGCG, all suspended in H2O. (Ex is the excitation wavelength, Em is the Emission Wavelength).
The above is our revised responses based on suggestions of the editor and reviewers.
Thank you again for your suggestions. We hope we can learn more from you.
We look forward to your reply!
Sincerely,
Prof. Na
Harbin University of Commerce

Reviewer 3 Report
I write regarding manuscript # foods-1268419 entitled " Hemp (Cannabis sativa L.) Seed Protein–Epigallocatechin Gallate Conjugates: Covalent Bonding and Functional Characteristics" submitted to the foods.
The authors need to follow the following instructions to improve this manuscript.
- Is hemp legal? Is commodity hemp production country legalized? Please add this information in your introduction.
- The title should change: The authors should skip the scientific name in the title. Instead, please use the well-known English name.
- Page 1, Line 19-23 (Abstract): Please concise and rewrite.
- Page 1, Line 39 (Introduction): Hemp (Cannabis sativa) is a native crop of China. Is it a crop?
- Please add replication numbers in each table and figure.
- Conclusion (Line 415-416): The authors should clarify.
- The authors used 64 references—too many. Please try to reduce (if possible) irrelevant and old ones.
- The authors should check grammar by a native speaker/professional editing company.
- Reference writing style should check.
- The authors should follow the author's guidelines.
Author Response
30-Jun-2021
Dear Editors and Reviewers:
Thank you for your letter and the comments on reviewers concerning our manuscript entitled “Hemp (Cannabis sativa L.) seed protein–EGCG conjugates: Covalent bonding and functional research” (ID: foods-1268419). Those comments are all valuable and extremely helpful for revising and improving our paper, and they also have an important guiding significance on our researches. We have studied your comments carefully and made corrections and revised portions are marked by red . We sincerely hope this manuscript will be finally acceptable to be published on the Foods.
Main corrections in the paper and the responses to the reviewer’s comments are as follows:
Responses to the reviewer’s comments:
Reviewer #3:
1 Is hemp legal? Is commodity hemp production country legalized? Please add this information in your introduction.
Response: Thank you for pointing out this problem. A 0.3% 9-tetrahydrocannabinol (THC) standard has been established by the European Union for this classification. If the THC is less than 0.3% is industrial hemp.it is allowed to be grown in China and Canada.Corresponding revisions have been made in the revised manuscript, please refer to the red of Line 46-48 in the revised manuscript.
2 The title should change: The authors should skip the scientific name in the title. Instead, please use the well-known English name.
Response: Appreciated your advice very much. According to your suggestion, the title has been revised. The revised title was " Hemp (Cannabis sativa L.) seed protein–EGCG conjugates: Covalent bonding and functional research "
3 Page 1, Line 19-23 (Abstract): Please concise and rewrite.
Response: Thank you for your suggestions, all of which are very important. We revised the content of Abstract and carried out a conciseand kept the mainly propose of this manuscript. The revised Abstract was shown on Line 24-25 of the revised manuscript.
4 Page 1, Line 39 (Introduction): Hemp (Cannabis sativa) is a native crop of China. Is it a crop?
Response: Appreciated your advice very much. Hemp is a kind of crop. The earliest hemp fibers were used in clothing. With the deepening of hemp research, hemp now has a wide range of applications. It can be used in food health care, medicine, chemical industry and so on. Hemp seed oil extraction, hemp root production of a variety of biological raw materials, hemp leaf extraction medicine.
5 Please add replication numbers in each table and figure.
Response: Thank you for pointing out our negligence. We have supplemented the parallel data, please check the figure and table in revised manuscript.
6 Conclusion (Line 415-416): The authors should clarify.
Response: Thank you for pointing out this problem. The application prospects of EGCG and protein covalent complexes as emulsifiers, but there are few studies on covalent complexes, and the determination of related indicators in this article can guide the preparation and application of the complexes. Corresponding revisions have been made in the revised manuscript, please refer to the red of Line 509-510 in the revised manuscript.
7 The authors used 64 references—too many. Please try to reduce (if possible) irrelevant and old ones.
Response: Thank you for pointing out this problem. We have deleted the irrelevant and old references of the original manuscript. Currently there are 48 text references.
8 The authors should check grammar by a native
Response: Thank you for your recognition and advices of our research, and these suggestions are very important. We have revised this article carefully.
9 Reference writing style should check.
Response: Thank you for your recognition and advices of our research. Reference writing style has been checked.
10 The authors should follow the author's guidelines.
Response: Thank you for pointing out this problem. We have revised the format of the article in accordance with the author's guidelines.
The above is our revised responses based on suggestions of the editor and reviewers.
Thank you again for your suggestions. We hope we can learn more from you.
We look forward to your reply!
Sincerely,
Prof. Na
Harbin University of Commerce
